# Prevalence and determinants of occupational injuries among emergency medical technicians in Northern Ghana

Ali Baba Awini[1], Douglas Aninng Opoku[2,3]*, Nana Kwame Ayisi-Boateng[4],
Joseph Osarfo[5], Alhassan Sulemana[6], Isaac Kofi Yankson[7], Maxwell Osei-Ampofo[4],
Ahmed Nuhu Zackaria[8], Sam Newton[9]

1 Ghana National Ambulance Service, Upper East Region, Ghana, 2 Department of Occupational and
Environmental Health, School of Public Health, Kwame Nkrumah University of Science and Technology,
Kumasi, Ghana, 3 Allen Clinic, Family Healthcare Services, Kumasi, Ghana, 4 Department of Medicine,
School of Medicine and Dentistry, Kwame Nkrumah University of Science and Technology, Kumasi, Ghana,
5 Department of Community Medicine, School of Medicine, University of Health and Allied Health Science,
Ho, Ghana, 6 Department of Environmental Science, College of Science, Kwame Nkrumah University of
Science and Technology, Kumasi, Ghana, 7 Council for Scientific and Industrial Research-Building and Road
Research Institute, Kumasi, Ghana, 8 Ghana National Ambulance Service, Headquarters, Accra, Ghana,
9 Department of Global and International Health, School of Public Health, Kwame Nkrumah University of
Science and Technology, Kumasi, Ghana

* douglasopokuaninng@gmail.com

University and Alex Ekwueme Federal University
Teaching Hospital, NIGERIA

**Data Availability Statement:** All relevant data are
within the paper and its Supporting Information
files.

## Abstract

### Background

Emergency Medical Technicians (EMTs) are the primary providers of prehospital emergency medical services. The operations of EMTs increase their risks of being exposed to occupational injuries. However, there is a paucity of data on the prevalence of occupational injuries among EMTs in sub-Saharan Africa. This study, therefore, sought to estimate the prevalence and determinants of occupational injuries among EMTs in the northern part of Ghana.

### Methods

A cross-sectional study was conducted among 154 randomly recruited EMTs in the northern part of Ghana. A pre-tested structured questionnaire was used to collect data on participants' demographic characteristics, facility-related factors, personal protective equipment use, and occupational injuries. Binary and multivariate logistic regression analyses with a backward stepwise approach were used to examine the determinants of occupational injuries among EMTs.

### Results

In the 12 months preceding data collection, the prevalence of occupational injuries among EMTs was 38.6%. Bruises (51.8%), and sprains/strains (14.3%) were the major types of injuries reported among the EMTs. The key determinants of occupational injury among EMTs were male sex (AOR: 3.39, 95%CI: 1.41–8.17), an absence of a health and safety

**Funding:** The authors received no specific funding for this work.

**Competing interests:** The authors have declared that no competing interests exist.

committee at the workplace (AOR: 3.92, 95%CI: 1.63–9.43), absence of health and safety policy at the workplace (AOR: 2.76, 95%CI: 1.26–6.04) and dissatisfaction with health and safety measures at the workplace (AOR: 2.51, 95%CI: 1.10–5.71).

## Conclusion

In the twelve months before to the data collection for this study, the prevalence of occupational injuries among EMTs of the Ghana National Ambulance Service was high. The creation of health and safety committees, the creation of health and safety rules, and the strengthening of current health and safety procedures for EMTs are all possible ways to lessen this.

## Introduction

Emergency Medical Technicians (EMTs) are prehospital experts who take care of acutely ill or injured patients and transport them timeously to and from health facilities. EMTs assess and oversee patients' medical treatment outside the hospital setting [1–4]. They contribute significantly to a nation's healthcare continuum of care by responding to calls concerning individual patients, major disasters and mass casualty incidents. The nature of their occupation exposes them to hazards, such as road traffic accidents en route to a scene, extreme temperatures, infectious agents, noise and long working hours which can increase their risks of occupational injuries and accidents. These can include travelling in light-and-siren-filled situations, exposure to violence, shift work, lifting heavy equipment and handling patients [1,5].

In the United States, data released by the National Institute for Occupational Safety and Health showed that approximately 75,400 emergency medical service workers sought care for non-fatal occupational injuries at emergency departments between 2017 and 2020 [6]. Similarly, out of a total of 21,749 cases of occupational injuries that were recorded in the USA among EMTs, 99.7% (21,690) of them resulted in non-fatality which accounted for lost work days [7]. Studies in the US and Australia have reported musculoskeletal injuries such as sprains and strains resulting from manual lifting and moving of patients as the commonest occupational injuries among emergency medical service workers [4,8,9]. Occupational claims in Australia between 2003 and 2012 have been reported to be between 4.8 to 6.3 times higher among ambulance officers compared to other occupations [10]. Motor vehicle accidents, needle stick injuries, and sharp injuries have been reported by Yilmaz et al in a study as the commonest mechanism of occupational injuries among EMTs in western Turkey [11]. In South Africa, Mcdowall and Laher in a study reported that over 26.3% of emergency medical service personnel sustained needle stick injuries [12].

Factors such as temporal employment, male gender, old age, long working hours, extreme temperatures, high-speed driving, lack of personal protective equipment (PPE) and poor safety culture have been reported to be associated with occupational injuries [13–18]. It has been reported that EMTs' exposure to conditions such as exposure to violence, shift work and lifting heavy equipment can increase their risk of occupational injuries compared to the entire population [1,19]. A study in the USA by Benavides et al reported that temporal workers had about two times increased risk of fatal injuries compared to permanent workers [16]. Another study in the USA by Reichard et al also reported that EMTs that were 45 years and above had about two times increased risk of occupational-related fatality compared to those who were 24 years and below [20]. One possible factor that can increase the risk of occupational injury is poor

adherence to PPE use at the workplace [21]. The use of PPE (such as safety boots, reflective jackets or vests, helmets, goggles, nose masks and hand gloves) can offer protection to workers against any work-related injuries, accidents and illnesses.

Over the past decade, the Ghana National Ambulance Service has undergone a massive transformation since its establishment in 2004 in Ghana. Consequently, EMTs have also received standardized training and updates over this period. They receive training on basic and advanced prehospital emergency care based on their scope of practice [22]. In Ghana, pre-hospital personnel can offer basic, advanced or critical emergency medical services [23]. Some studies in Ghana have assessed occupational injuries [24,25], and burnout [26,27] among healthcare workers (such as nurses, physicians and laboratory technicians). However, there is no available data on the prevalence and determinants of occupational injuries among EMTs of the Ghana National Ambulance Service. Data is required to determine the most important and cost-effective measures that could be made to enhance the health and safety of EMTs of the Ghana National Ambulance Service. This study, therefore, sought to estimate the prevalence and determinants of occupational injuries among EMTs of the Ghana National Ambulance Service. It provides baseline data on the overall prevalence and determinants of occupational injuries among EMTs, which can inform policy direction to enhance the health and safety of EMTs in the Ghana National Ambulance Service. Also, this study could serve as a reference in the monitoring and evaluation of interventions that may be rolled out to address injuries in this population.

## Methods

### Study design and setting

A cross-sectional study was conducted among EMTs of the Ghana National Ambulance Service in three (Upper East, Upper West and Northern Regions) out of five regions in the northern part of Ghana from 30th October 2020 to 1st December 2020.

In 2008, the Ghana National Ambulance Service began providing emergency medical care (including transporting patients to and from health facilities) in these regions from just one station each, which was situated in Bolgatanga, Wa, and Tamale. The Ghana National Ambulance Service now has stations in all the regions in the northern part of Ghana, as well as a regional dispatch centre and regional administration [28].

The EMTs of the Ghana National Ambulance Service provide services such as the provision of pre-hospital emergency medical care to accident victims by transporting them from road traffic crash scenes to a health facility, standby emergency cover at mass public gatherings and also liaise with other emergency care providers in time of disaster or mass casualties to provide pre-hospital care to victims [28]. The 2022 annual report of the Ghana National Ambulance Service from the Upper East, Upper West and Northern Regions of Ghana showed that the three regions conducted 5,797 cases of hospital referrals run with an average monthly number of 483 cases in 2022. In total, there are forty-six ambulances in the three regions (Upper East: 16, Upper West: 12 and Northern Region: 18).

### Study population and eligibility criteria

The study population were all EMTs of the Ghana National Ambulance Service in the three regions in the northern part of Ghana. In total, there were two hundred and fifty-three (253) EMTs of the Ghana National Ambulance Service in the three regions (Upper East: 89, Upper West: 69 and Northern Region: 95). All EMTs who have had at least twelve months of working experience with the Ghana National Ambulance Service were included in the study. All EMTs

of the Ghana National Ambulance Service that were on their annual leave at the time of the study were excluded.

## Sample size determination

The sample size for this study was estimated using Cochran's formula for sample size calculation: *Sample size*, $SS = \frac{Z^2 pq}{d^2}$, where 'Z' is the chi-square degree of freedom value in the table at a 95% confidence level (1.96), 'p' is the population proportion (29.7% of the proportion of occupational injuries sustained among health workers in Ghana [25]), q is (1 –p) and 'd' is the degree of accuracy expressed as a proportion (0.05). The sample size was estimated as follows:

$$Sample\ size, SS = \frac{1.96^2 \text{ x } 0.297 \text{ x } (1 - 0.297)}{0.05^2} = 321 \qquad \text{Eq1}$$

The sample size was adjusted to 142 using the finite population correction formula for the study using Eq 2:

$$New\ sample\ size, n = \frac{SS}{1 + \frac{SS-1}{N}} \qquad \text{Eq2}$$

where n is the new sample size, SS is the estimated sample size, 321, and N is the finite population, 253 (which is the total number of EMTs in the three regions). Using a non-response rate of 10.0%, a total of 156 EMTs were recruited for the study. The prevalence of occupational injuries among healthcare workers in Ghana [25] was used for the sample size estimation because there is no previous study on the injury rate among EMTs in Ghana. Also, EMTs are part of the healthcare workers in Ghana. Again, the dynamics and work environment in which EMTs operate in developed countries are very different from Ghana.

## Sampling procedure

First, three regions were selected at random from five regions in the northern part of Ghana. Proportional allocation of the sample size to the size of the target population was used to select the number of EMTs from each region. In total, 55, 43 and 58 EMTs were selected from Upper East, Upper West, and Northern regions respectively. The staff list in each region was retrieved from all the regional administration offices. Staff were assigned codes to identify them and this was used for balloting by an independent person to select the participants for each region. The EMTs that were selected were contacted on phone and scheduled appointments for the questionnaire administration.

## Data collection instrument and process

A pre-tested structured questionnaire was used to collect demographic factors (age, sex, years of work, marital status, level of education), facility-related factors (availability of health and safety committee and policy, health and safety training, provision of PPE) and occupational injuries (types, contributory factors, location of accidents). The questionnaire was self-designed de novo by the authors through a literature review [7,11,15,20,29]; it was not validated. However, the questionnaire was pretested among fifteen EMT officers in the Ashanti Region. This region is located in the central part of Ghana and is largely cosmopolitan with varied sociodemographic characteristics. All the necessary corrections were made to the data collection instrument to improve its reliability and precision after the pretest before using it to collect the study data. Occupational injury in this study was defined as any injury or accident that occurred through the performance of duty at the workplace [30]. The data was gathered

using a self-administered approach among study participants. In each region, two research assistants were trained by the investigators on ethics in research and obtaining consent from research participants. The research assistants contacted the EMTs on phone to schedule an appointment for the questionnaire administration. For participants that had sustained more than one occupational injury in the last twelve months, data on their most recent injury was captured.

## Data management and statistical analysis

Data entry was done in Epi Info version 7.2.4.0 (Centres for Disease Control and Prevention [CDC], USA). The data was cleaned and checked for correctness and consistency before analysis. Analysis was performed using Stata version 16 (StataCorp, College Station, USA) statistical software package. Means with standard deviation and medians with interquartile range were used to describe continuous variables while frequencies and percentages were used to describe categorical variables. Univariate logistic regression analysis was used to obtain Crude Odds Ratio (COR) with statistical significance considered at p-value < 0.05. A multivariate logistic regression analysis using a backward stepwise approach (using a p of 0.05) was used to examine the independent determinants of occupational injuries. All variables including non-statistically significant independent variables in the univariate logistic regression were deemed important from previous studies [9,11,15,31]. Hence, we adopted a backward stepwise approach and included all of them in the multivariate analysis. Statistical significance was maintained at a p-value of 0.05 with a 95% confidence interval for the final multivariate model. The primary outcome of this study was the prevalence of occupational injuries in the twelve months preceding the data collection. Prevalence of occupational injuries was calculated as the proportion of EMTs that reported sustaining an occupational injury in the twelve months preceding the data collection out of all EMTs enrolled into the study and expressed as a percentage.

## Ethics approval and consent to participate

Ethical approval was granted by the Committee on Human Research, Publications and Ethics, School of Medicine and Dentistry, Kwame Nkrumah University of Science and Technology, Kumasi, Ghana with reference number CHRPE/AP/403/20. Study participants were briefed about the study objectives, risks and benefits of participation. They were encouraged to ask questions for clarification. All study participants signed a written informed consent before recruitment.

## Results

### Demographic characteristics of study participants

Of 156 EMTs that were recruited for the study, 154 of them completed the questionnaire with a response rate of 98.7%. The mean age of study participants was 35.5 years (±7.1) with a range of 20 to 53 years. Approximately 73.0% of the 154 study participants were males and close to two-thirds (64.9%) had secondary education. The median years of work was 5 years (interquartile range: 6). Approximately 76.0% of the study participants were in the operations department (Table 1).

### Occupational injuries and personal protective equipment usage among study participants

Over a 12-month period, the prevalence of occupational injuries among the EMTs was 38.6%. Out of the 59 injured, over half (51.8%) were bruises and about 39.7% of the injuries occurred

**Table 1. Demographic characteristics of study participants.**

| Variables | Frequency | Percentage, % [Range] |
|---|---|---|
| Age group (years) | | |
| 20–29 | 34 | 22.1 |
| 30–39 | 79 | 51.3 |
| ≥ 40 | 41 | 26.6 |
| Mean (±SD) | 35.5 (±7.1) | [20 – 53] |
| Sex | | |
| Male | 112 | 72.7 |
| Female | 42 | 27.3 |
| Level of education | | |
| Basic | 5 | 3.3 |
| Secondary | 100 | 64.9 |
| Tertiary | 49 | 31.8 |
| Marital status | | |
| Single | 59 | 38.3 |
| Married | 95 | 61.7 |
| Years of work (years) | | |
| 1–5 | 95 | 61.7 |
| 6–10 | 40 | 26.0 |
| ≥ 11 | 19 | 12.3 |
| Median years of work (IQR) | 5 (6) | [1 – 16] |
| Department | | |
| Administration | 12 | 7.7 |
| Dispatch/control | 25 | 16.3 |
| Operations | 117 | 76.0 |

IQR: Interquartile range

SD: Standard deviation.

at a road traffic accident scene. Over half (56.9%) of those injured reported a lack of adequate PPE as a factor that contributed to its occurrence. Approximately 70.1% of the EMTs reported that the PPE supplied to them was adequate. About two-thirds (65.6%) of the EMTs studied reported that they were satisfied with the health and safety measures at their workplace. Again, over 71.4% and 52.0% of the EMTs indicated the presence of health and safety committees and policies at their workplaces respectively (Table 2).

## Determinants of occupational injuries among study participants

After adjusting for significant covariates in the stepwise logistic regression, male EMTs (AOR: 3.39, 95%CI: 1.41–8.17), an absence of a health and safety committee at the workplace (AOR: 3.92, 95%CI: 1.63–9.43), absence of health and safety policy at the workplace (AOR: 2.76, 95% CI: 1.26–6.04) and dissatisfaction with health and safety measures at the workplace (AOR: 2.51, 95%CI: 1.10–5.71) were associated with occupational injury among EMTs (Table 3).

## Discussion

The prevalence of occupational injury among study participants in the 12 months preceding data collection was 38.6%. The 36.8% prevalence rate observed in this study underpins an expanding body of evidence that EMTs are at an increased risk of occupational injuries

**Table 2. Occupational injury, type, causes and PPE use among study participants.**

| Variables | Frequency | Percentage, % |
|---|---|---|
| Experienced injury in the past 12 months | | |
| Yes | 59 | 38.6 |
| No | 94 | 61.4 |
| Type of injury (n = 56) | | |
| Fracture | 6 | 10.7 |
| Bruises | 29 | 51.8 |
| Sprain & strain | 8 | 14.3 |
| Laceration/burns | 6 | 10.7 |
| Others* | 7 | 12.5 |
| Location of the accident (n = 58) | | |
| Road traffic accident scene | 23 | 39.7 |
| Hospital | 8 | 13.8 |
| Office | 8 | 13.8 |
| Residence of a patient | 2 | 3.5 |
| In the ambulance | 17 | 29.3 |
| Factors affecting injuries occurrence (n = 58) | | |
| Inadequate training in health and safety | 5 | 8.6 |
| Inadequate PPE | 33 | 56.9 |
| Lack of knowledge of health and safety issues by the employee | 11 | 19.0 |
| Not sure | 9 | 15.5 |
| Report injury/accident (n = 59) | | |
| Yes | 31 | 52.5 |
| No | 28 | 47.5 |
| Provision of PPE | | |
| Adequate | 108 | 70.1 |
| Inadequate | 46 | 29.9 |
| Usage of PPE at the workplace | | |
| Sometimes | 49 | 31.8 |
| Always | 105 | 68.2 |
| Health and safety committee at the workplace | | |
| Yes | 110 | 71.4 |
| No | 44 | 28.6 |
| Health and safety policy at the workplace | | |
| Yes | 80 | 52.0 |
| No | 74 | 48.0 |
| Satisfaction with health and safety measures at the workplace | | |
| Satisfied | 101 | 65.6 |
| Not satisfied | 53 | 34.4 |

PPE: Personal protective equipment.

*Includes needle stick injuriesinjuries from electrical shock, dislocation.

[4,7,29,32]. The nature of their work involves bending, carrying and lifting patients [33–35] which can increase their risks of sustaining injuries. A study among 246 healthcare workers in Ghana reported a 12-month occupational injury prevalence of 29.7% [25]. Also, the injury rate (36.8%) observed in this study is similar to the prevalence of occupational injury (38.8%) reported among paramedics in a study in South Africa [18]. Reichard et al reported an injury

**Table 3. Factors influencing occupational injuries among study participants.**

| Variable | Injury n (%) | Crude OR 95%CI | P -value | Adjusted OR 95%CI | P -value |
|---|---|---|---|---|---|
| Age group (years) | | | | | |
| 20–29 | 9 (26.5) | 1.00 | | - | - |
| 30–39 | 32 (40.5) | 1.89 (0.78–4.58) | 0.158 | - | - |
| ≥ 40 | 18 (43.9) | 2.17 (0.82–5.79) | 0.121 | - | - |
| Sex | | | | | |
| Male | 50 (44.6) | 2.96 (1.29–6.75) | 0.010 | 3.39 (1.41–8.17) | 0.029 |
| Female | 9 (21.4) | 1.00 | | 1.00 | |
| Level of education | | | | | |
| Basic | 3 (60.0) | 1.00 | | - | |
| Secondary | 33 (33.0) | 0.33 (0.05–2.06) | 0.235 | - | - |
| Tertiary | 23 (46.9) | 0.59 (0.09–3.85) | 0.581 | - | - |
| Marital status | | | | | |
| Single | 21 (35.6) | 1.00 | | - | - |
| Married | 38 (40.0) | 1.21 (0.62–2.36) | 0.585 | - | - |
| Years of work (years) | | | | | |
| 1–5 | 28 (29.5) | 1.00 | | - | - |
| 6–10 | 19 (47.5) | 2.16 (1.01–4.64) | 0.047 | - | - |
| ≥ 11 | 12 (63.2) | 4.16 (1.46–11.50) | 0.007 | - | - |
| Department | | | | | |
| Administration | 6 (50.0) | 1.00 | | - | - |
| Dispatch/control | 6 (24.0) | 0.32 (0.07–1.36) | 0.121 | - | - |
| Operations | 47 (40.2) | 0.67 (0.20–2.21) | 0.512 | - | - |
| Health and safety committee at the workplace | | | | | |
| Yes | 38 (34.6) | 1.00 | | 1.00 | |
| No | 21 (47.7) | 1.73 (0.85–3.52) | 0.130 | 3.92 (1.63–9.43) | 0.002 |
| Health and safety policy at the workplace | | | | | |
| Yes | 23 (28.8) | 1.00 | | 1.00 | |
| No | 36 (48.7) | 2.35 (1.21–4.57) | 0.012 | 2.76 (1.26–6.04) | 0.011 |
| Provision of PPE | | | | | |
| Adequate | 41 (38.0) | 1.00 | | - | - |
| Inadequate | 18 (39.1) | 1.05 (0.52–2.13) | 0.892 | - | - |
| Usage of PPE at the workplace | | | | | |
| Sometimes | 20 (40.8) | 1.00 | | - | - |
| Always | 39 (37.1) | 0.86 (0.43–1.71) | 0.662 | - | - |
| Satisfaction with health and safety measures at the workplace | | | | | |
| Satisfied | 32 (31.7) | 1.00 | | 1.00 | |
| Not satisfied | 27 (50.9) | 2.24 (1.13–4.43) | 0.021 | 2.51 (1.10–5.71) | 0.029 |

OR: Odds ratio.

rate of 8.6 per 100 full-time equivalents among emergency medical service workers in the USA [4]. Maguire et al also in a retrospective analysis of data on all occupational injuries reported an injury rate of 34.6 per 100 full-time workers per year in the USA [36]. Yilmaz et al reported an incidence rate of injuries of 10.9 among EMTs in Turkey [11]. However, all these studies [4,11,36] reported incidence rates which makes it difficult to directly compare to the prevalence rate in the present study. Reichard et al utilized records of all occupational injuries collected through the National Electronic Injury Surveillance System from 2010 to 2014 in the

USA which can account for the high incidence rate of occupational injuries in their study [4]. Similarly, the high incidence rate of occupational injuries reported by Maguire et al USA [36] could also be attributed to the use of injury records from two urban agencies' registries on all reported cases of occupational injuries from 1998 to 2002 in the USA. In Ghana, there is no such system in place that keeps records of all occupational injuries. The commonest type of occupational injury (bruises, sprains and strains) observed among EMTs in the present study is similar to what was reported in studies among EMTs in the US [7,37].

The present study observed that lack of adequate PPE (56.9%) was a major contributory factor to the occurrence of occupational injuries. The significance of PPE use in injury prevention has been discussed in the literature [38,39]. The health and safety of EMTs depend on high adherence to PPE use which can make the difference between safety and injuries at the workplace. The use of PPE can reduce the level of exposure to hazards when both engineering and administrative controls are not practical or efficient to minimize the risk of exposure to the barest minimum. In California, a study by Mathews et al reported that one of the barriers that hinder adherence to PPE use among paramedics is inadequate access to safety devices [40]. This study highlights the need to ensure there is an adequate supply of PPE for use to reduce the prevalence of occupational injuries among EMTs in Ghana.

We observed in the present study that male EMTs had about three times (AOR: 3.39, 95% CI: 1.41–8.17) increased odds of sustaining occupational injury compared to their female counterparts. A reason may be that males are often involved in tasks which may demand the use of sophisticated machines that can predispose them to high risks of occupational injuries, unlike females whose tasks may be less demanding (such as administrative, and dispatch/control) which reduces their exposure risks and subsequent occurrence of occupational injuries. This is consistent with reports in the US by the Bureau of Labor Statistics that men suffer more occupational injuries than women [41].

Health and safety committees work closely with management and employees in identifying potential workplace hazards and prevent them from causing accidents, illnesses and injuries at the workplace [42]. It was observed in the present study that in the absence of a health and safety committee at the workplace, EMTs had about four times (AOR: 3.92, 95%CI: 1.63–9.43) increased risks of sustaining occupational injury compared to those that had it at their workplace. This is consistent with reports in Canada and the USA that a health and safety committee is associated with lower injury rates at the workplace [43,44]. The absence of a health and safety committee at the workplace can affect an organization's ability to routinely conduct a risk assessment which is important for identifying potential sources of hazards. Health and safety committees provide supervisory support to workers which can help to mitigate injury rates. Inadequate supervisory support has been reported by Yanar et al in a study in Canada to be significantly associated with an increased risk of physical injuries at the workplace [31].

It was observed in this study that EMTs that did not have a health and safety policy at the workplace had about three times (AOR: 2.76, 95%CI: 1.26–6.04) increased odds of sustaining occupational injury compared to those that had it. A possible reason could be that those without a health and safety policy may not have a good safety culture (like the use of PPE, and safety training at the workplace) and are also not likely to have a health and safety committee which is a significant strategy for injury prevention [21,45]. The existence of a health and safety policy indicates an institution's willingness to promote employee health and safety. Thus, it is in every organization's best interest to have and implement health and safety policies and measures to reduce workplace illnesses, accidents and injuries. In the United Kingdom (The Health Safety Work Act 1974 and the Management of the Health and Safety at Work Regulations 1999), it is mandatory for companies to provide health and safety policies for their staff [46,47] unlike Ghana where there is no such law.

A significant finding of this study is the approximately three times (AOR: 2.51, 95%CI: 1.10–5.71) increased odds of occupational injuries among EMTS that were dissatisfied with health and safety measures at the workplace compared with those that were satisfied. This may be explained by the fact that their dissatisfaction with health and safety may negatively affect their commitment to adhering to safety procedures and measures at the workplace which can increase their risk of sustaining an occupational injury. When workers perceive health and safety systems to be positive at the workplace, their level of adherence to safety regulations and measures increases [48].

## Strengths and limitations of the study

This is the first study to estimate the prevalence and determinants of occupational injuries among EMTs in Ghana and provides useful data for future studies. A strength of this study is the representativeness of the study population in the Northern part of Ghana which enhances the generalizability of study findings. The EMTs were selected from all the regional branches of the Ghana National Ambulance Service in three regions in the Northern part of Ghana. This study provides useful data for strategies to improve the health and safety of EMTs of the Ghana National Ambulance Service.

The study has some limitations. Participants had to recall occupational injuries that had occurred in the 12 months prior to data collection and it is possible recall bias may have led to some underestimation of the prevalence of occupational injuries among the EMTs. Neverthe-less, the prevalence and types of occupational injuries reported are comparable to findings from previous studies [17,18,37]. Secondly, the use of cross-sectional design limits our ability to examine the causes of occupational injuries among EMTs. Lastly, although the data collec-tion instrument was not validated, pretesting was conducted to improve its reliability and validity.

## Conclusion

In the twelve months before to the data collection for this study, the prevalence of occupational injuries among EMTs of the Ghana National Ambulance Service was high. The commonest occupational injuries sustained were bruises, sprains and strains. The occurrence of occupa-tional injuries in this study was influenced by factors such as male EMTs, an absence of a health and safety committee, the absence of health and safety policies and dissatisfaction with health and safety measures at the workplace. The creation of health and safety committees, the creation of health and safety rules, and strengthening current health and safety procedures for EMTs are all possible ways to lessen this.

## Supporting information

**S1 Dataset.**
(DTA)

## Acknowledgments

We are grateful to the EMTs, management and staff of the Ghana National Ambulance Service in the Upper East, Upper West and Northern regions of Ghana. We also express our profound gratitude to all the research assistants who assisted with the recruitment of study participants.

## Author Contributions

**Conceptualization:** Ali Baba Awini, Douglas Aninng Opoku, Sam Newton.

**Data curation:** Ali Baba Awini, Douglas Aninng Opoku, Nana Kwame Ayisi-Boateng, Joseph Osarfo, Isaac Kofi Yankson, Ahmed Nuhu Zackaria.

**Formal analysis:** Douglas Aninng Opoku.

**Methodology:** Ali Baba Awini, Douglas Aninng Opoku, Nana Kwame Ayisi-Boateng, Joseph Osarfo, Alhassan Sulemana, Isaac Kofi Yankson, Maxwell Osei-Ampofo, Sam Newton.

**Project administration:** Douglas Aninng Opoku, Nana Kwame Ayisi-Boateng, Joseph Osarfo, Alhassan Sulemana, Ahmed Nuhu Zackaria.

**Supervision:** Sam Newton.

**Validation:** Ali Baba Awini, Douglas Aninng Opoku, Joseph Osarfo, Alhassan Sulemana, Isaac Kofi Yankson, Maxwell Osei-Ampofo, Ahmed Nuhu Zackaria, Sam Newton.

**Writing – original draft:** Douglas Aninng Opoku, Nana Kwame Ayisi-Boateng, Joseph Osarfo, Alhassan Sulemana, Maxwell Osei-Ampofo.

**Writing – review & editing:** Ali Baba Awini, Douglas Aninng Opoku, Nana Kwame Ayisi-Boateng, Joseph Osarfo, Alhassan Sulemana, Isaac Kofi Yankson, Maxwell Osei-Ampofo, Ahmed Nuhu Zackaria, Sam Newton.

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
