## [Decision Letter · Decision Letter 0]

9 Mar 2023

PONE-D-23-04075Occupational injuries among Emergency Medical Technicians: Burden and determinants in Northern Ghana

PLOS ONE

Dear Dr. Opoku,

Thank you for submitting your manuscript to PLOS ONE. After careful consideration, we feel that it has merit but does not fully meet PLOS ONE publication criteria as it currently stands. Therefore, we invite you to submit a revised version of the manuscript that addresses the points raised during the review process.

For acceptance for publication, you are required to pay special attention to the third publication criterion of PLOS ONE which addresses statistics and analyses. Both reviewers have raised these concerns in their comments on the methodology section. Please explain in clearer terms how you measured both prevalence and incidence (you stated the prevalence and incidence). I recommend you also address the other issues raised by both reviews. The manuscript will benefit from copy-editing.

Please include the following items when submitting your revised manuscript: A rebuttal letter that responds to each point raised by the academic editor and reviewers. You should upload this letter as a separate file labeled Response to Reviewers; A marked-up copy of your manuscript that highlights changes made to the original version. You should upload this as a separate file labelled Revised Manuscript with Track Changes.

An unmarked version of your revised paper without tracked changes. You should upload this as a separate file labelled Manuscript. If applicable, we recommend that you deposit your laboratory protocols in protocols.io to enhance the reproducibility of your results. Protocols.io assigns your protocol its own identifier (DOI) so that it can be cited independently in the future. For instructions see: https://journals.plos.org/plosone/s/submission-guidelines#loc-laboratory-protocols. Additionally, PLOS ONE offers an option for publishing peer-reviewed Lab Protocol articles, which describe protocols hosted on protocols.io. Read more information on sharing protocols at https://plos.org/protocols?utm_medium=editorial-email&utm_source=authorletters&utm_campaign=protocols.

We look forward to receiving your revised manuscript.

Kind regards,

Adaoha Pearl Agu, MBBS, MSc, FMCPH

Academic Editor PLOS ONE

Reviewers' comments:

Reviewer's Responses to Questions

**Comments to the Author**

1. Is the manuscript technically sound, and do the data support the conclusions?

Reviewer #1: Yes

Reviewer #2: Yes

2. Has the statistical analysis been performed appropriately and rigorously? 

Reviewer #1: No

Reviewer #2: Yes

3. Have the authors made all data underlying the findings in their manuscript fully available?

Reviewer #1: Yes

Reviewer #2: Yes

4. Is the manuscript presented in an intelligible fashion and written in standard English?

Reviewer #1: Yes

Reviewer #2: Yes

5. Review Comments to the Author

Reviewer #1: S.noComponents My Comments and Suggestions

1.Title •It is coherent and brief but, Northern Ghana is large areas, please specific the exact area/s

2.Abstract•Methods: It is well constructed, but what about why didn’t include the binary logistic regression analysis in method section?

•Result: (aOR….) correct through your document including abstract as (AOR….)

•Conclusion: do not start with statement “Approximately three out of ten EMTs reported an occupational injury within a 49 twelve-month period”. Please revise the conclusion part

•

3.IntroductionWell organized

4.Statements of the Problems Well organized

5.Objective Well convey

6.Methods•Setting: Inconsistency usage of study area; i.e. in your title and objective, you used it Northern Ghana, but in your methods, it said (Upper East: 89, Upper West: 69 and 136 Northern regions: 95). Please modified appropriately.

•Sample size estimation and sampling:

1.Change as Sample size Determination

2.I think you have two objectives

3.Objective: Prevalence of Occ. Injuries

4.Objective 2: Associated Factors of Occ. Injuries . So, did you calculate the sample size for associated factors? You should calculate it and compare with objective 1, and large objective will be your sample size

•Data Quality: To insure your quality of data was/were done?

1.Tools used for assessment

2.Language clarity

3.Reliability and validity?

4. where is your pretested conducted? please address these

•Data analysis:

1.How you got CRUDE ODD Ratio if you didn’t use binary logistic regression.

2.indicate the candidate variables/assumed to factors/ those moved to multivariate logistic regression

3.What was the

modeled in the multivariate logistic regression analysis? 0.25 or 0.20 p-value please indicate again

•Ethical consideration: Use in place Journal formal guideline

7.Result•Good, but it need to modify the titles of each table considering

1.what?

2.Where?

3.Who?

4.When?

5.Example

Table 1 Demographic characteristics of study participants, so here you only indicated what and who….

•Table 3: Determinants of occupational injuries among study participants: use AOR rather than (aOR: 3.92, 95%CI: 1.63 – 9.43)

•If you used backward step

8.Discussion•Good, some coherent paragraphs were used, please recheck it

9.Conclusion •Modify, do not conclude with numerical or percentages [38.6%]

•[may use sever, high mild or low] prevalence of occupational problem/injuries is found among………………, which need………………….

10.References •DOI is required for those have

11.General Comments •Good research areas and article, but it need major revision.

Reviewer #2: Thank you for completing this important research. Such research is the necessary precursor to efforts to mitigate the problem of EMTs around the world suffering from high rates of occupational injuries. I have made some suggested edits for your consideration. Please keep up this important work.

6. PLOS authors have the option to publish the peer review history of their article (what does this mean?). If published, this will include your full peer review and any attached files.

Reviewer #1: No

Reviewer #2: **Yes: **Brian J Maguire

---

## [Author Response · Author response to Decision Letter 0]

16 Mar 2023

Academic Editor

Academic Editor’s comments: Both reviewers have raised these concerns in their comments on the methodology section. Please explain in clearer terms how you measured both prevalence and incidence (you stated the prevalence and incidence). I recommend you also address the other issues raised by both reviews. The manuscript will benefit from copy-editing.

Authors’ response; Thank you very much for this. All concerns that were raised by the reviewers have been addressed. We did not consider incidence and this has been expunged from the manuscript. The revised section now reads as;

…..The primary outcome of this study was the prevalence of occupational injuries in the twelve months preceding the data collection. Prevalence of occupational injuries was calculated as the proportion of EMTs that reported sustaining an occupational injury in the twelve months preceding the data collection out of all EMTs enrolled into the study and expressed as a percentage

Reviewer 1

Reviewer’s comments; It is coherent and brief but, Northern Ghana is large areas, please specific the exact area/s

Authors’ response; We are grateful to the reviewer for his comment on the study title. However, we respectfully would like to maintain the current study title. This is because this study was conducted in three out of five regions in the northern part of Ghana. In our title, ‘Northern’ is being used to denote the geographical area of Ghana and not the political or administrative area. We have clarified it in the study setting in the methods.

Reviewer’s comments; Methods: It is well constructed, but what about why didn’t include the binary logistic regression analysis in method section?

Authors’ response; This has been done as suggested by the reviewer. The section in reference is revised to read as;

…..Binary and multivariate logistic regression analysis with a backward stepwise approach was used to examine the determinants of occupational injury among EMTs.

Reviewer’s comments; Result: (aOR….) correct through your document including abstract as (AOR….)

Authors’ response; All ‘aOR’ in the manuscript have been changed to ‘AOR’ as suggested by the reviewer.

Reviewer’s comments; Conclusion: do not start with statement “Approximately three out of ten EMTs reported an occupational injury within a 49 twelve-month period”. Please revise the conclusion part

Authors’ response; This has been revised as suggested. The section in reference is revised to read as;

….The burden of occupational injuries among EMTs of the Ghana National Ambulance Service within a 12-month period prior to data collection in this study was high

Reviewer’s comments; Setting: Inconsistency usage of study area; i.e. in your title and objective, you used it Northern Ghana, but in your methods, it said (Upper East: 89, Upper West: 69 and 136 Northern regions: 95).

Authors’ response; Upper East, Upper West and Northern regions are three out of five regions in Ghana. These regions have been created for administrative or political purposes. The study was conducted in three (Upper East, Upper West and Northern regions) randomly selected out of the five regions in northern Ghana. This has been clarified in the revised manuscript. We use ‘northern Ghana’ in its geographical sense. The section in reference is revised to read as;

….. A cross-sectional study was conducted among EMTs of the Ghana National Ambulance Service in three (Upper East, Upper West and Northern Regions) out of five regions in the Northern part of Ghana from 30th October 2020 to 1st December 2020.

…… First, three regions were selected at random from five regions in the northern part of Ghana.

Reviewer’s comments; Please modified appropriately. Sample size estimation and sampling: Change as Sample size Determination

Authors’ response; This has been done as suggested. Thank you

Reviewer’s comments; I think you have two objectives: 1. Objective: Prevalence of Occ. Injuries Objective 2: Associated Factors of Occ. Injuries. So, did you calculate the sample size for associated factors? You should calculate it and compare with objective 1, and large objective will be your sample size

Authors’ response; Firstly, we used the prevalence to calculate the sample size because it was the primary outcome in the study. Secondly, the sample size used was reasonable even for the associated factors if one considers that most of the 95% CI around the Odds Ratios were not unreasonably wide. This suggests reasonable precision from a reasonable sample size.

Reviewer’s comments; Data Quality: To ensure your quality of data was/were done? Tools used for assessment., language clarity, reliability and validity?

Authors’ response; The data collection instrument was developed de novo by the authors through literature review and was not validated. However, to improve on its reliability and validity, it was pretested on 15 EMTs in the Ashanti Region. This has been acknowledged as a limitation in the revised manuscript. Again, data quality checks were performed by the authors by checking for correctness and consistency. 

Reviewer’s comments; where is your pretested conducted? please address these 

Authors’ response; It was conducted in the Ashanti Region. This is a densely populated region located in the central part of Ghana. This region is largely cosmopolitan with varied sociodemographic characteristics. This has been acknowledged in the manuscript. 

Reviewer’s comments; Data analysis: How you got CRUDE ODD Ratio if you didn’t use binary logistic regression. indicate the candidate variables/assumed to factors/ those moved to multivariate logistic regression. What was the modeled in the multivariate logistic regression analysis? 0.25 or 0.20 p-value please indicate again 

Authors’ response; This has been explained in the revised manuscript. The revised section now reads as;

…..Univariate logistic regression analysis was used to obtain Crude Odds Ratio (COR) with statistical significance considered at p-value < 0.05. A multivariate logistic regression analysis using a backward stepwise approach (using a p of 0.05) was used to examine the independent determinants of occupational injuries. All variables including non-statistically significant independent variables in the univariate logistic regression were deemed important from previous studies (10,12,16,31). Hence, we adopted a backward stepwise approach and included all of them in the multivariate analysis. Statistical significance was maintained at a p-value < 0.05 with a 95% confidence interval for the final multivariate model.

Reviewer’s comments; Ethical consideration: Use in place Journal formal guideline 

Authors’ response; This has been revised to ‘Ethics approval and consent to participate’

Reviewer’s comments; Good, but it need to modify the titles of each table considering what? Where? Who? When? Example Table 1 Demographic characteristics of study participants, so here you only indicated what and who….

Authors’ response; We appreciate the comments of the reviewer on the naming of the titles of the tables. However, the authors would like to maintain the current titles of all the tables as they briefly describe the contents of all the tables in the context of the study. This is also in line with the journal’s requirements. Besides, there is a good chance of being repetitive.

Reviewer’s comments; Table 3: Determinants of occupational injuries among study participants: use AOR rather than (aOR: 3.92, 95%CI: 1.63 – 9.43)

Authors’ response; This has been done as suggested. 

Reviewer’s comments; If you used backward step

Authors’ response; A backward stepwise approach was used and this has been acknowledged in the manuscript.

Reviewer’s comments; Good, some coherent paragraphs were used, please recheck it 

Authors’ response; Thank you. This has been done as suggested

Reviewer’s comments; Modify, do not conclude with numerical or percentages [38.6%] [may use sever, high mild or low] prevalence of occupational problem/injuries is found among………………, which need………………….

Authors’ response; This has been done as suggested. The revised section now reads;

…..The burden of occupational injuries among the EMTs of the Ghana National Ambulance Service in this study was high

Reviewer’s comments; DOI is required for those have

Authors’ response; The DOI have been provided as recommended.

Reviewer’s comments; Good research areas and article, but it need major revision.

Authors’ response; Thank you for your comments. The manuscript has been revised based on reviewers’ comments. We are very grateful. 

Reviewer 2

Reviewer’s comments; ….(EMTs) are the primary providers of….

Authors’ response; This has been done as suggested 

Reviewer’s comments; … However, there is a paucity of data on the burden of occupational injuries among EMTs in sub-Saharan Africa This study… place a ‘full stop’ after Africa

Authors’ response; This has been done. 

Reviewer’s comments; You seem to be using the words prevalence and incidence interchangeably. Your methods show that you are focused on incidence. 

Authors’ response; The main outcome of the work was to estimate the prevalence of occupational injury. We have reported only prevalence in the revised manuscript. Thank you. 

The revised section now reads as;

…..The primary outcome of this study was the prevalence of occupational injuries in the twelve months preceding the data collection. Prevalence of occupational injuries was calculated as the proportion of EMTs that reported sustaining an occupational injury in the twelve months preceding the data collection out of all EMTs enrolled into the study and expressed as a percentage

Reviewer’s comments; ‘Protection’ is an unclear word in this context. Instead, I recommend that you report the AOR for men. 

Authors’ response; This has been done as suggested. We have presented the AOR for men as suggested in the revised manuscript. 

Reviewer’s comments; The main outcome of this study was a self-reported occupational injury sustained in the last twelve months before the data collection (highlighted).

Authors’ response; This has been deleted from the data collection instrument and process section as highlighted

Reviewer’s comments; Define what 6 means in the bracket…. The median years of work was 5 years (6)

Authors’ response; This has been defined as suggested. The revised section now reads;

The median years of work was 5 years (interquartile range: 6).

Reviewer’s comments; ‘Labor’ instead of ‘labour’

Authors’ response; This has been corrected as suggested 

Reviewer’s comments; EMTs had about ‘four’ and not ‘three’ times (aOR: 3.92, 95%CI: 1.63 – 275 9.43) 

Authors’ response; This has been corrected as suggested 

Reviewer’s comments; Add ‘a’ between that and health….. in Canada and the USA that health and safety….

Authors’ response; This has been done as suggested. 

Reviewer’s comments; ….had about two times (aOR: 2.76, 95%CI: 1.26 – 6.04) Usually round to the closest whole number. So "three" here.

Authors’ response; This has been done as suggested

Reviewer’s comments; Ideally add the name of the university and, if possible, a URL. Also, for this type of reference a URL is ideally included.

Authors’ response; This has been done as suggested

---

## [Decision Letter · Decision Letter 1]

27 Mar 2023

PONE-D-23-04075R1Occupational injuries among Emergency Medical Technicians: Burden and determinants in Northern GhanaPLOS ONE

Dear Dr. Opoku,

Thank you for submitting your manuscript to PLOS ONE. After careful consideration, we feel that it has merit but does not fully meet PLOS ONE’s publication criteria as it currently stands. Therefore, we invite you to submit a revised version of the manuscript that addresses the points raised during the review process. 

The attempt at revision is a good one. Appropriate statistics is part of Publication criteria 3 for PLOS ONE. The reason for my decision of a major revision is because of the sample size formula you used for calculation of sample size. That formula is for small sample techniques and some of the assumptions in that formula don’t reflect general opinion. One of the main purposes of sample size calculations for estimating prevalence is representativeness. Up and unless the calculated sample size is bigger than the known finite source population, there is no need to apply a constraining formula like the Krejcie formula. I request that you attempt a re-calculation with the Cochrane formula in determining the sample size for one proportion: n=(Zsq x PQ) /dsq. Since you used a sampling method other than simple random sampling, there is need to account for the dilution of variability. This is done by multiplying the above formula with a design effect value (usually>=1.5). It is not necessary to compute a design effect here so you can assume a design effect value from 1.5 and above.  If after recalculation with varying assumptions, you do not get a sample size that works; the study will benefit from putting a cautionary statement in the discussion that reflects this.

Your title should be changed to reflect the view of the reviewer. However replace “burden” with “prevalence” in the title and all through the text since you did not actually deal with burden.

Please attend to the other comments of the reviewer.

We look forward to receiving your revised manuscript.

Kind regards,

Adaoha Pearl Pearl Agu, MBBS, MSc, FMCPH

Academic Editor

PLOS ONE

Reviewers' comments:

Reviewer's Responses to Questions

**Comments to the Author**

1. If the authors have adequately addressed your comments raised in a previous round of review and you feel that this manuscript is now acceptable for publication, you may indicate that here to bypass the “Comments to the Author” section, enter your conflict of interest statement in the “Confidential to Editor” section, and submit your "Accept" recommendation.

Reviewer #1: (No Response)

Reviewer #2: All comments have been addressed

2. Is the manuscript technically sound, and do the data support the conclusions?

Reviewer #1: (No Response)

Reviewer #2: (No Response)

3. Has the statistical analysis been performed appropriately and rigorously? 

Reviewer #1: Yes

Reviewer #2: (No Response)

4. Have the authors made all data underlying the findings in their manuscript fully available?

Reviewer #1: Yes

Reviewer #2: (No Response)

5. Is the manuscript presented in an intelligible fashion and written in standard English?

Reviewer #1: Yes

Reviewer #2: (No Response)

6. Review Comments to the Author

Reviewer #1: Title

I prefer it if your title will be modified as “Burden and determinants of Occupational injuries among Emergency Medical Technicians in Northern Ghana”

Abstract

1. Introduction…Good

2. Objective…Good

3. Method…Good

4. Result …..good

5. Conclusion

• Still I’m not satisfied with the conclusion, hence need rewrite again

I prefer it if your title will be modified as “In the twelve months before to the data collection for this study, the burden of occupational injuries among EMTs of the Ghana National Ambulance Service was high. The creation of health and safety committees, the creation of health and safety rules, and strengthening current health and safety procedures for EMTs are all possible ways to lessen this”

6. Keywords: Should be arranged in alphabetical, it is not seeking a long phrase, it is about words, thus correct each long phrase used for MeSH, Revise it again

Introduction

• Please use risks of occupational injuries mentioned line#66 should be moved to risk factors found between line #84-#94. Minor revision of introduction

Methods

• Source of information under study area?

• Line #115-131 should be cited

• Sample size calculation: Use standard sample size calculation among “Equation” found on the task word

• What does “This explains why a ‘P’ from a developed country was not used?” mean line 153-154

Result

• I suggesting you please avoid each bold of variables (such as Age group, sex,… found Table 1 , Experienced injury in the past 12 months…etc in Table 2, associated factors (Age, education……mentioned in Table 3

Discussion

• Inappropriate, line 257-259: The key determinants………..

• unnecessarily statement, line 259-261: To the best to……..

• I strongly advise you to compare and contrast the current main finding with other studies with potential discrepancies once you have mentioned the current main finding

Conclusion

• Still I’m not satisfied with the conclusion, hence need rewrite again

Reviewer #2: (No Response)

7. PLOS authors have the option to publish the peer review history of their article (what does this mean?). If published, this will include your full peer review and any attached files.

Reviewer #1: **Yes: **Sina Temesgen Tolera

Reviewer #2: **Yes: **Brian J. Maguire

---

## [Author Response · Author response to Decision Letter 1]

8 Apr 2023

Academic Editor

Academic Editor’s comments: The reason for my decision of a major revision is because of the sample size formula you used for calculation of sample size. That formula is for small sample techniques and some of the assumptions in that formula don’t reflect general opinion. One of the main purposes of sample size calculations for estimating prevalence is representativeness. Up and unless the calculated sample size is bigger than the known finite source population, there is no need to apply a constraining formula like the Krejcie formula. I request that you attempt a re-calculation with the Cochrane formula in determining the sample size for one proportion: n = (Zsq x PQ) /dsq. Since you used a sampling method other than simple random sampling, there is need to account for the dilution of variability. This is done by multiplying the above formula with a design effect value (usually>=1.5). It is not necessary to compute a design effect here so you can assume a design effect value from 1.5 and above. If after recalculation with varying assumptions, you do not get a sample size that works; the study will benefit from putting a cautionary statement in the discussion that reflects this.

Authors’ response; We have used the Cochrane formula in determining the sample size for one proportion as suggested. We further adjusted the sample size using the finite population correction formula. This approach has been used in other studies [1-4]. The revised section on the sample size estimation now reads;

The sample size for this study was estimated using Cochran’s formula for sample size calculation: Sample size,SS= (Z^2 pq)/d^2 , where ‘Z’ is the chi-square degree of freedom value in the table at a 95% confidence level (1.96), ‘p’ is the population proportion (29.7% of the proportion of occupational injuries sustained among health workers in Ghana (25)), q is (1 – p) and ‘d’ is the degree of accuracy expressed as a proportion (0.05). The sample size was estimated as follows:

Sample size,SS= (〖1.96〗^2 x 0.297 x (1-0.297))/〖0.05〗^2 =321 Equation 1

The sample size was adjusted to 142 using the finite population correction formula for the study using Equation 2:

New sample size,n= SS/(1+ (SS-1)/N ) Equation 2

where n is the new sample size, SS is the estimated sample size, 321, and N is the finite population, 253 (which is the total number of EMTs in the three regions). Using a non-response rate of 10.0%, a total of 156 EMTs were recruited for the study. The prevalence of occupational injuries among healthcare workers in Ghana (25) was used for the sample size estimation because there is no previous study on the injury rate among EMTs in Ghana. Also, EMTs are part of the healthcare workers in Ghana. Again, the dynamics and work environment in which EMTs operate in developed countries are very different from Ghana.

References 

 Agu AP, Umeokonkwo C, Adeke A, Nnabu CR, Ossai E, Azuogu B. Awareness of occupational hazards, use of personal protective equipment and workplace risk assessment among welders in mechanic village, Abakaliki, Southeast Nigeria: Awareness of occupational hazards among welders. Nigerian Medical Journal. 2021;62(3):113-21.

 Bentum L, Brobbey LK, Adjei RO, Osei-Tutu P. Awareness of occupational hazards, and attitudes and practices towards the use of personal protective equipment among informal woodworkers: the case of the Sokoban Wood Village in Ghana. International Journal of Occupational Safety and Ergonomics. 2022 Jul 3;28(3):1690-8.

 Mbaisi EM, Wanzala P, Omolo J. Prevalence and factors associated with percutaneous injuries and splash exposures among health-care workers in a provincial hospital, Kenya, 2010. Pan African Medical Journal. 2013 Apr 29;14(1).

 Ofori AA, Osarfo J, Agbeno EK, Manu DO, Amoah E. Psychological impact of COVID-19 on health workers in Ghana: A multicentre, cross-sectional study. SAGE Open Medicine. 2021 Mar;9:20503121211000919.

Academic Editor’s comments: Your title should be changed to reflect the view of the reviewer. However, replace “burden” with “prevalence” in the title and all through the text since you did not actually deal with burden

Authors’ response; This has been done as suggested. All “burden” in the manuscript have been replaced with “prevalence” The revised title now reads as;

Prevalence and determinants of occupational injuries among Emergency Medical Technicians in Northern Ghana

Reviewer 1

Reviewer’s comments; I prefer it if your title will be modified as “Burden and determinants of Occupational injuries among Emergency Medical Technicians in Northern Ghana”

Authors’ response; The study title has been revised based on your comment and that of the Academic Editor. 

The revised title now reads as;

Prevalence and determinants of occupational injuries among Emergency Medical Technicians in Northern Ghana

Reviewer’s comments; Methods: Good

Authors’ response; Thank you

Reviewer’s comments; Methods: good 

Authors’ response; Thank you

Reviewer’s comments; Results: good 

Authors’ response; Thank you

Reviewer’s comments; Still I’m not satisfied with the conclusion, hence need rewrite again. I prefer it if your title will be modified as “In the twelve months before to the data collection for this study, the burden of occupational injuries among EMTs of the Ghana National Ambulance Service was high. The creation of health and safety committees, the creation of health and safety rules, and strengthening current health and safety procedures for EMTs are all possible ways to lessen this”

Authors’ response; This has been revised as suggested. The section in reference is revised to read as;

…..In the twelve months before to the data collection for this study, the prevalence of occupational injuries among EMTs of the Ghana National Ambulance Service was high. The creation of health and safety committees, the creation of health and safety rules, and strengthening current health and safety procedures for EMTs are all possible ways to lessen this.

Reviewer’s comments; Keywords: Should be arranged in alphabetical, it is not seeking a long phrase, it is about words, thus correct each long phrase used for MeSH, Revise it again 

Authors’ response; This has been revised as suggested.

Reviewer’s comments; Please use risks of occupational injuries mentioned line#66 should be moved to risk factors found between line #84-#94. Minor revision of introduction 

Authors’ response; This has been done as suggested. Thank you

Reviewer’s comments; Setting: Source of information under study area? Line #115-131 should be cited

Authors’ response; This has been done as suggested 

Reviewer’s comments; Sample size calculation: Use standard sample size calculation for on (S = 𝑋 2 NP (1 – P) ÷ 𝑑 2 (N – 1) + 𝑋 2 P (1 – P) among “equations” found on the task word like the following style, it is not your, but do in same fashion. 

S=(x^2 (NP(1-P))/(d^2 (N-1))+⋯

Authors’ response; We have presented the sample size equation in the fashion recommended by the reviewer

Reviewer’s comments; What does “This explains why a ‘P’ from a developed country was not used?” mean line 153-154 

Authors’ response; This statement has been removed from the revised manuscript

Reviewer’s comments; I suggesting you please avoid each bold of variables (such as Age group, sex,… found Table 1 , Experienced injury in the past 12 months…etc in Table 2, associated factors (Age, education……mentioned in Table 3

Authors’ response; This has been done as suggested. Thank you.

Reviewer’s comments; Inappropriate, line 257-259: The key determinants………..

Authors’ response; This has been removed from the revised manuscript. 

Reviewer’s comments; unnecessarily statement, line 259-261: To the best to……..

Authors’ response; This has been removed from the revised manuscript. 

Reviewer’s comments; I strongly advise you to compare and contrast the current main finding with other studies with potential discrepancies once you have mentioned the current main finding

Authors’ response; This has been done as suggested. We could not find many prevalence studies on EMTs injuries to compare our findings. The studies that have been conducted about the subject estimated incidence of injury rate per full-time equivalence which makes it difficult for direct comparison since we estimated prevalence. The revised section now reads;

…. Reichard et al reported an injury rate of 8.6 per 100 full-time equivalents among emergency medical service workers in the USA [4]. Maguire et al also in a retrospective analysis of data on all occupational injuries reported an injury rate of 34.6 per 100 full-time workers per year in the USA [36]. Yilmaz et al reported an incidence rate of injuries of 10.9 among EMTs in Turkey [11]. However, all these studies [4, 11, 36] reported incidence rates which makes it difficult to directly compare to the prevalence rate in the present study. Reichard et al utilized records of all occupational injuries collected through the National Electronic Injury Surveillance System from 2010 to 2014 in the USA which can account for the high incidence rate of occupational injuries in their study [4]. Similarly, the high incidence rate of occupational injuries reported by Maguire et al USA [36] could also be attributed to the use of injury records from two urban agencies' registries on all reported cases of occupational injuries from 1998 to 2002 in the USA. In Ghana, there is no such system in place that keeps records of all occupational injuries.

Reviewer’s comments; Still I’m not satisfied with the conclusion, hence need rewrite again

Authors’ response; The conclusion has been revised as suggested. Thank you. The section in reference now reads as;

In the twelve months before to the data collection for this study, the prevalence of occupational injuries among EMTs of the Ghana National Ambulance Service was high. The commonest occupational injuries sustained were bruises, sprains and strains. The occurrence of occupational injuries in this study was influenced by factors such as male EMTs, an absence of a health and safety committee, the absence of health and safety policies and dissatisfaction with health and safety measures at the workplace. The creation of health and safety committees, the creation of health and safety rules, and strengthening current health and safety procedures for EMTs are all possible ways to lessen this.

---

## [Editor Report · Decision Letter 2]

12 Apr 2023

Prevalence and determinants of occupational injuries among Emergency Medical Technicians in Northern Ghana

PONE-D-23-04075R2

Dear Dr., Opoku,

We’re pleased to inform you that your manuscript has been judged scientifically suitable for publication and will be formally accepted for publication once it meets all outstanding technical requirements.

Kind regards,

Adaoha Pearl Agu, MBBS, MSc, FMCPH

Academic Editor

PLOS ONE
---

## [Editor Report · Acceptance letter]

17 Apr 2023

PONE-D-23-04075R2 

Prevalence and determinants of occupational injuries among Emergency Medical Technicians in Northern Ghana 

Dear Dr. Opoku:

I'm pleased to inform you that your manuscript has been deemed suitable for publication in PLOS ONE. Congratulations! Your manuscript is now with our production department. 

Kind regards, 

on behalf of

Dr. Adaoha Pearl Pearl Agu 

Academic Editor

PLOS ONE